# Immune Pathways with Aging Characteristics Improve Immunotherapy Benefits and Drug Prediction in Human Cancer

**DOI:** 10.3390/cancers15020342

**Published:** 2023-01-04

**Authors:** Xinyue Wang, Shuang Guo, Hanxiao Zhou, Yue Sun, Jing Gan, Yakun Zhang, Wen Zheng, Caiyu Zhang, Xiaoxi Zhao, Jiebin Xiao, Li Wang, Yue Gao, Shangwei Ning

**Affiliations:** College of Bioinformatics Science and Technology, Harbin Medical University, Harbin 150081, China

**Keywords:** pan-cancer, aging, immunotherapy, prognosis, biomarker

## Abstract

**Simple Summary:**

By integrating large-scale transcriptomic and genomic data, this study identified immune-related pathways with aging characteristics among 25 cancer types. We found that the perturbation of immune-related pathways showed a cancer-specific expression pattern. Moreover, the aging- and cancer-related dysregulated patterns of pathways were in the same direction in most cancer types. Furthermore, the study found that aging-related immune pathways with clinical relevance improved immunotherapy benefits in melanoma and non-small cell lung cancer patients. Using a network-based method, this study predicted potential drug targets for immunotherapy and treatment combinations. Collectively, this study may provide insight into cancer immunotherapy, thus improving treatment benefits.

**Abstract:**

(1) Background: Perturbation of immune-related pathways can make substantial contributions to cancer. However, whether and how the aging process affects immune-related pathways during tumorigenesis remains largely unexplored. (2) Methods: Here, we comprehensively investigated the immune-related genes and pathways among 25 cancer types using genomic and transcriptomic data. (3) Results: We identified several pathways that showed aging-related characteristics in various cancers, further validated by conventional aging-related gene sets. Genomic analysis revealed high mutation burdens in cytokines and cytokines receptors pathways, which were strongly correlated with aging in diverse cancers. Moreover, immune-related pathways were found to be favorable prognostic factors in melanoma. Furthermore, the expression level of these pathways had close associations with patient response to immune checkpoint blockade therapy in melanoma and non-small cell lung cancer. Applying a net-work-based method, we predicted immune- and aging-related genes in pan-cancer and utilized these genes for potential immunotherapy drug discovery. Mapping drug target data to our top-ranked genes identified potential drug targets, FYN, JUN, and SRC. (4) Conclusions: Taken together, our systematic study helped interpret the associations among immune-related pathways, aging, and cancer and could serve as a resource for promoting clinical treatment.

## 1. Introduction

Evidence from accumulating studies has substantiated that aging is one of the most fundamental causes of a broad range of cancer [1,2,3]. The tumor incidence and mortality risk increase dramatically during aging [4]. Another well-established hallmark of cancer is the perturbation of the immune system [5,6,7]. Aging of the immune system leads to the impairment of both innate and adaptive immune processes in the body, which can create an inflammatory environment [8,9,10]. An inflamed tumor microenvironment could thus influence the immune landscape [11]. 

Immunotherapy has proven a promising strategy for cancer treatment, for which impressive results have been achieved in various cancers [12]. By inhibiting checkpoint molecules, such as programmed cell death 1 (PD-1), programmed cell death 1 ligand 1 (PD-L1), and cytotoxic T-lymphocyte associated protein 4 (CTLA-4), patients have achieved durable benefits after immune checkpoint blockade (ICB) therapies [13,14]. The main targets of ICB therapy are T cells, which can construct and maintain the immune landscape [15,16]. The decline of the immune system during the aging process could lead to the dysfunction of T cells; thus, patients do not respond to ICB therapy [17]. Current studies from a multi-omic view have identified biomarkers related to immunotherapy response, such as the expression of PD-L/PD-L1 [18,19]. However, rather than being isolated from each other, genes often interact closely when they function [20]. The role of the immune-related pathway in aging and cancer remains elusive. Comprehensively characterizing the role of the immune-related pathway in aging and cancer might help better understand immunotherapy resistance mechanisms.

We integrated genomic and transcriptomic data of 25 cancer types to refine these gaps further. In this study, we systematically analyzed the age- and cancer-associated differences in immune-related pathways in pan-cancer. We noticed that the perturbation of immune-related pathways during aging showed cancer-specific characteristics. Besides, the age-related alteration patterns of these pathways were investigated. We identified several immune-related pathways that could predict the patient response to immunotherapy in melanoma and non-small cell lung cancer. Potential drug targets of ICB therapies were predicted by a network-based method. Collectively, our results might help elucidate the age-associated roles of immune-related pathways in tumorigenesis and possibly inform immunotherapy options.

## 2. Materials and Methods

### 2.1. Data Collection

The transcripts per million (TPM) of RNA-seq data for 25 cancer types from the Cancer Genome Atlas data portal (TCGA) and non-cancerous tissues from Genotype-Tissue Expression (GTEx) consortium were downloaded from the UCSC Xena platform [21] (http://xena.ucsc.edu/, accessed on 19 November 2020) (Appendix A). TCGA and GTEx transcript expression were integrated using UCSC TOIL pipeline [22], and 19,725 protein-coding genes were retained. Clinical and phenotype data were also downloaded from the UCSC Xena platform.

Cancer types included for further analysis were as follows: adrenocortical carcinoma (ACC), bladder urothelial carcinoma (BLCA), breast invasive carcinoma (BRCA), cervical squamous cell carcinoma and endocervical adenocarcinoma (CESC), colon adenocarcinoma (COAD), esophageal carcinoma (ESCA), glioblastoma multiforme (GBM), kidney chromophobe (KICH), kidney renal clear cell carcinoma (KIRC), kidney renal papillary cell carcinoma (KIRP), brain lower grade glioma (LGG), liver hepatocellular carcinoma (LIHC), lung adenocarcinoma (LUAD), lung squamous cell carcinoma (LUSC), ovarian serous cystadenocarcinoma (OV), pancreatic adenocarcinoma (PAAD), prostate adenocarcinoma (PRAD), rectum adenocarcinoma (READ), skin cutaneous melanoma (SKCM), stomach adenocarcinoma (STAD), testicular germ cell tumors (TGCT), thyroid carcinoma (THCA), thymoma (THYM), uterine corpus endometrial carcinoma (UCEC) and uterine carcinosarcoma (UCS).

Half maximal inhibitory concentration (IC50) value of drugs in tumor cell lines was obtained from Genomics of Drug Sensitivity in Cancer (GDSC) database [23] (https://www.cancerrxgene.org/, accessed on 22 May 2022). Gene expression profile of cancer cell lines was downloaded from Cancer Cell Line Encyclopedia (CCLE) database (https://sites.broadinstitute.org/ccle, accessed on 22 May 2022) (Appendix A).

### 2.2. Identification of Dysregulated IRPs in 25 Cancer Types

Seventeen immune-related pathways including 1793 genes were obtained from the Immunology Database and Analysis Portal (ImmPort) project [24] (http://www.immport.org, accessed on 11 March 2021). We applied single-sample gene-set enrichment analysis (ssGSEA) to calculate the immune-related pathways (IRP) scores across 25 cancer types based on ImmPort genesets (Appendix A), IRP scores were scaled by MinMax normalization to the interval [0,1] by the following equation:(1)$$X_{normalized}=\frac{X-X_{min}}{X_{max}-X_{min}}$$
where *X* is each IRP score across samples.

(1)Identification of differentially expressed IRPs in tumor samples

R package ‘limma’ was applied to extract dysregulated IRPs in tumor samples [25]. IRPs with Benjamini-Hochberg (BH) adjusted *p*-value < 0.05 were considered as differentially expressed.

(2)Identification of IRPs differentially expressed with age

It must be mentioned that the actual age of samples (donors from whom the sample was collected) from GTEx was not provided; thus, we approximated the median for each age group to each sample, i.e., 25 for all samples from 20–29 (Appendix A). A linear regression model generated by limma was used to extract IRPs differentially expressed with age [26,27]. IRPs with a *p*-value < 0.05 was determined as statistically significant. Moreover, we applied Analysis of Variance (ANOVA) analysis to compare the IRP scores differences across cancers.

### 2.3. Benchmark Genesets to Evaluate the Aging Characteristics of IRPs

Aging-related-genesets (ARG) were downloaded from Molecular Signatures Database [28] (https://www.gsea-msigdb.org/gsea/msigdb, accessed on 7 May 2022), Human Aging Genomic Resources [29] (http://genomics.senescence.info/genes/, accessed on 7 May 2022) and a previously published study [30]. We calculated ARG scores using the ssGSEA method based on the above aging-related genesets. We applied Spearman’s rank correlation to assess the relationship between IRPs and ARGs at the pan-cancer and cancer-specific levels. Besides, we calculated mutual information (R package ‘MICMIC’ [31]) between IRPs and ARGs to reinforce the results obtained from Spearman’s analysis. 

### 2.4. Mutation Analysis

The somatic mutation data of the mutation annotation format (MAF) file was downloaded from the publicly available TCGA MC3 project [32]. IRP alteration frequencies were computed as the fraction of altered samples across cancers. Besides, we calculated the tumor mutation burden (TMB) for each sample by R package ‘maftools’ [33]. We applied Spearman’s rank correlation analysis to assess the relations between IRP scores and TMB.

### 2.5. Survival Analysis

We performed the Cox regression model and log-rank tests to identify IRPs related to overall survival in 25 cancer types. Besides, the multivariate-Cox regression model was performed in SKCM to identify IRPs that served as independent prognostic factors. KM survival curve grouped by the median of IRP scores were plotted by R package ‘survival’ and ‘survminer’. We constructed a nomogram by R package ‘rms’ to select independent prognostic factors and evaluated the predictive capability of nomograms by the calibration plot.

### 2.6. Prediction of Immunotherapy Response of IRPs

Five cohorts of SKCM patients received immunotherapy, including GSE91061, GSE96619, GSE35640, Gide dataset, and VanAllen dataset [34,35,36,37,38], and two cohorts of (non-small-cell lung cancer) NSCLC patients after ICB therapy, including GSE135222 and GSE126044, were obtained from previous studies [39,40]. Patients who completely or partially responded to the treatment were considered responders (Appendix A). IRP scores in these datasets were calculated by the ssGSEA method among protein-coding genes. In the current study, we evaluated the predictive ability of IRP scores using two methods: (1) The Area Under Curve (AUCs) from Receiver operating characteristic curve (ROC) curve analysis for discriminating responders vs. non-responders by R package ‘ROCR’; (2) Wilcoxon test (and multi-Wilcoxon test) in patients with different responses after immunotherapy (*p* < 0.05 was considered significant). Moreover, we used the Gide cohort as the training set and applied the support vector machines (SVM) model to predict the response of patients from the TCGA SKCM cohort. We used the R package e1071 to train the SVM model.

### 2.7. Network-Based Method Identifying Immune-Related Genes as Drug Targets

A protein–protein interaction network, including 141,296 interactions among 13,460 genes, was compiled from a previous study [41]. 1509 immune-related genes (IRGs) in 17 IRPs were obtained from the ImmPort project [24]. First, we identified IRGs that were both dysregulated in tumors and with age in 25 cancers (same procedure as identification of dysregulated IRPs), then the rank of each IRG in a specific cancer type was defined as
(2)$$R_{IMG}=\sqrt{R_{\left|FC_{1}\right|}\times R_{\left|FC_{2}\right|}}$$
where *FC*_1_ is the fold change of IRG in tumor and *FC*_2_ is the fold change with age.

After obtaining a ranked list of IRGs, we projected the top 10% of genes to the PPI network in 25 cancers, separately. A random walk with a restart was employed to measure the association between all other genes in the network and the top 10% IRGs by the following equation [42,43]:(3)$$P_{t+1}=\gamma P_{0}+\left(1-\gamma \right)WP_{t}$$
where *P*_0_ is the initial probability vector in which “1” were assigned to the top 10% IRGs; *γ* is the restarting probability; *W* denotes the adjacency matrix of the network; *P_t_* is the final probability vector of other nodes after iteratively reaching stability. 

For each cancer type, genes that score in the top 1% after network propagation were extracted, then we integrated the top genes shared among 25 cancers and re-performed a random walk with a restart. Finally, we obtained a ranked gene list for drug discovery by the Drug Gene Interaction Database (DGIdb) database [44] (https://www.dgidb.org/, accessed on 12 June 2022). Furthermore, we used the drug sensitivity data from GDSC and CCLE to validate our drug prediction results.

### 2.8. Functional Enrichment Analysis

Functional enrichment analysis consists of Kyoto Encyclopedia of Genes and Genomes (KEGG) pathways, Reactome genesets and Wikipathways was carried out using Metascape platform [45] (https://metascape.org/, accessed on 17 June 2022).

## 3. Results

### 3.1. Disturbance of Immune-Related Pathways between Normal and Tumor Tissues among 25 Cancer Types

We first performed the ssGSEA method to calculate immune-related pathways (IRP) scores in 25 cancer types using the RNA-seq profile from the TCGA and GTEx database. Then we explored the differential expression of these IRPs between tumor and normal samples. We found that most IRPs were dysregulated in more than 19 cancer types (Figure 1A), except the Interferons pathway. Almost all cancer types displayed complex expression patterns of IRPs; however, all IRPs were significantly up-regulated in TGCT and down-regulated in lung cancers, suggesting a distinct immune landscape in different cancer types (Appendix A). IRP scores were highly positively correlated with each other at the pan-cancer level, suggesting a close interaction of various immune pathways (Figure 1B). Next, we compared the IRP scores in tumor samples among 25 cancer types. We found that most IRP scores varied dramatically in different cancer types (ANOVA, *p* < 0.05), suggesting cancer-type specific immune mechanisms (Figure 1C). In contrast, Cytokines, Interleukins, and TGF-β family member receptors showed a narrow range of scores among cancers (Appendix A). These specific expression patterns were further confirmed by uniform manifold approximation and projection (UMAP) analysis (Figure 1D). Cancers with related tissue origin were clustered together, exhibiting similar IRP expression patterns, including central nervous system cancer (LGG and GBM), genitourinary cancer (KIRC, KIRP, KICH, BLCA, PRAD, and TGCT), thoracic cancer (LUAD, LUSC), gynecologic cancer (OV, UCEC, CESC, and UCS) and Core gastrointestinal cancer (ESCA, STAD, COAD and READ). Taken together, IRPs showed cancer-specific characteristics, suggesting diverse roles in the oncogene process and tumor microenvironment.

### 3.2. Dysregulated Immune-Related Pathways Showed Age-Associated Characteristics among Cancers

As we have identified differentially expressed IRPs in tumor samples among 25 cancer types, we then assessed the association between patients’ age and IRP score in pan-cancer. Using a linear regression model, we obtained a list of IRPs that showed significantly differential expression with age (Figure 2A, Appendix A). The results were further validated by Spearman’s correlation rank and mutual information analysis (Appendix A). The strongest negative association was found in TGCT as all 17 IRPs were down-regulated with age in TGCT. On the other hand, none of the IRPs showed age-associated characteristics in KICH. Despite different expression patterns, all IRPs were dysregulated with age in at least ten cancer types (Figure 2A). Next, Fisher’s exact test was used to perform an overlap analysis between dysregulated tumor-IRPs and dysregulated age-IRPs in 25 cancer types (Figure 2B (right), Appendix A). Although 17 IRPs showed diverse changing directions between tumor and age across cancer types, significant overlaps found in most cancer types (led by BRCA) were IRPs changed in the same direction between tumor and age (Figure 2B,C). Results suggested a simultaneous molecular changing pattern between aging and tumor development. However, in LGG and THCA, the overlap was significant for IRPs changed in the opposite direction. Due to the small number of dysregulated tumor-IRPs and age-IRPs in several cancers, including ACC, CESC, and KIRC, no significant overlaps were found. In LGG, 13 in 17 IRPs presented increased expression with age and tumor. Different changing direction cases were found in READ, LUAD, and THYM, whereas in cancers like GBM, all IRPs showed only one changing direction with no significant overlaps (Figure 2C, Appendix A).

### 3.3. Benchmark Gene Sets Validated the Aging-Related Characteristics of IRPs

Several studies have collected essential genes associated with age. Here, we obtained ten gene sets from different sources to evaluate the aging-related value of 17 IRPs [28,29,30]. First, ssGSEA were used to calculate age-related gene sets (ARG) scores in both pan-cancer and cancer-specific level. Next, Spearman’s rank correlation analysis was used to calculate the correlation coefficient between IRP and ARG scores. We found that most IRP scores and ARG scores showed significantly positive correlations, both at the all- and tumor-sample level; the results were further validated by mutual information analysis (Figure 3A, Appendix A). Strikingly, we found that the IMM-AGE gene set showed the strongest positive correlation with most of the 17 IRPs (Rho > 0.3, *p* < 0.05) at all levels. IMM-AGE is a gene signature developed by multi-omic-based approaches to explore immune-aging dynamics at high resolution [30]. The high correlation values suggested that IRP was a meaningful metric for aging in tumorigenesis. In addition, HAGR and GO_AGING gene sets were also conventional aging benchmarks; our analysis showed that the TCR signaling pathway was highly correlated with the above gene sets at the pan-cancer level, suggesting potential roles of T cell receptors in aging (Figure 3B). Consistent with the pan-cancer analysis, the strongest positive correlations were found between IMM-AGE and TCR signaling pathway in almost all cancer types (Appendix A). Besides, HAGR and GO_AGING were also highly correlated with the TCR signaling pathway in cancer-specific analysis, including GBM, STAD, TGCT, and STAD. Taken together, our results might indicate that TCR and BCR signaling pathways could serve essential roles in aging and cancer (Figure 3C).

### 3.4. Cytokine Receptors Pathway Showed a High Alteration Frequency among 25 Cancer Types

Numerous studies have proved the essential role of tumor mutation burden in aging. During the aging process, accumulated somatic mutations were associated with increased or decreased risks of tumorigenesis. Besides, ongoing studies have revealed the relevance of TMB and immune checkpoint blockade (ICB) therapy response [46]. Herein, we calculated the sample alteration frequencies for each IRP in 25 cancer types (Figure 4A). Tumor samples with at least one alteration of pathway genes were considered altered samples. Results showed that the TCR signaling pathway, Cytokines, Cytokines receptors, and Antimicrobials had relatively high alteration frequencies across all cancer types (Figure 4D), which was consistent with previous studies [42]. Using logistic regression adjusting for cancer type, we found that 9 in 17 IRPs were positively correlated with age at the pan-cancer level; the strongest correlation was found between Cytokine receptors and age (Figure 4B). In cancer-specific analysis, all but Interferon-related pathways had significant correlations with at least one cancer type (Figure 4C). The strongest correlations were found in STAD, whereas mutations in 3 IRPs were more common in younger than older UCEC patients, which was partially due to the higher proportion of hypermutated tumors in young patients [26] (Figure 4C,E). On the other hand, negative correlations found in lung cancers might be partly due to the smoking status of young patients [26] (Figure 4C,E, Appendix A). Next, we computed the TMB of each tumor sample using maftools. Then we assessed the correlation between IRP score and TMB at a cancer-specific level by a linear regression model. Results showed complex relations between IRP scores and TMB in pan-cancer. Although 17 IRP scores correlated with sample TMBs in different directions across cancers, an overall negative correlation between multiple IRP scores and sample TMB was found across several cancer types, including GBM, LIHC, and THCA (Appendix A). In summary, these findings from a genomic view established a strong relation between immune-related pathways, aging, and cancer development.

### 3.5. IRPs Displayed a Strong Clinical Relevance among 25 Cancer Types

Given the significant associations between IRPs and aging in cancer, we thus investigate whether IRPs could serve as valuable markers with prognostic value. A univariate Cox regression model was used to evaluate the association between IRP score and overall survival in 25 cancer types. Results showed that all IRPs had clinical relevance in at least three cancer types (uni-Cox *p*-value < 0.05). 17 of 25 cancer types had at least one IRP associated with patient survival; among them, 15 out of 17 IRPs were favorable survival factors (HR < 1) in SKCM (Figure 5A). KM survival curve showed that high IRP scores were associated with better survival rates in SKCM (Figure 5B, Appendix A). Next, we applied a multivariate Cox regression model to test whether IRPs could serve as independent prognostic factors in SKCM. As a result, Antimicrobials, Interferons, Interferon receptors, and Interleukins pathway were independent OS prognostic factors in SKCM. It is worth noting that Interleukins pathway was a protective factor in the uni-Cox analysis, whereas becoming a risky factor of OS in multi-Cox analysis (Figure 5C). A nomogram was constructed based on independent prognostic factors to predict 3-year and 5-years overall survival probability and median survival time in SKCM (Figure 5D). The 3- and 5-year calibration plots showed that predicted survival probability by nomogram had a little deviation from actual survival probability (Figure 5E). Taken together, our results enforced that immune-related pathways had high prognostic values in pan-cancer.

### 3.6. IRP Scores Were Associated with Immunotherapy Response of Patients

There are clear associations between IRPs and tumor immunity; thus, we speculated that IRPs could serve as indicators for the response of patients with immunotherapy. In the current analysis, we assessed the predictive abilities of IRPs in immunotherapy response using patients with SKCM and NSCLC, for which we could obtain available cohorts as much as possible. Here, we collected five cohorts of SKCM patients (1 MAGE - A3 and 4 CTLA-4 and PD-1 blockade immunotherapy) and two cohorts of NSCLC patients who received PD-1 blockade immunotherapy. First, we computed the AUCs to evaluate the predictive ability of each IRP in discriminating between responder and non-responder in SKCM (Appendix A). As a result, six pathways, including Antigen Processing and Presentation, Natural Killer Cell Cytotoxicity, Antimicrobials, Interleukins Receptor, BCR Signaling, and TCR Signaling pathways, achieved AUCs greater than 0.7 in 5 independent cohorts (Figure 6B). Considering the small size of some cohorts, we additionally applied the Wilcoxon method to avoid the unreliability of the ROC curve. Consistent with the AUC result, these six IRP scores were significantly different between responder and non-responder in at least three cohorts (Figure 6A). 

Moreover, we used the Gide cohort to predict the immunotherapy response of TCGA SKCM patients using the SVM model; a prolonged overall survival was found in predicted responders (Figure 6C). We also found that SKCM patients with “immune” phenotype were prone to respond to ICB therapy [47] (Figure 6D). The expression level of the six above pathways was all highest in “immune” phenotype SKCM patients (Appendix A). Furthermore, we assessed the clinical implications of IRPs in patients after ICB therapy by the Gide cohort; results indicated that IRPs were significantly related to the OS and PFS of patients who received ICB therapy (Appendix A). Taken together, the results indicated that higher IRP scores might have an association with durable benefits for clinical immunotherapy.

Next, the same procedures were carried out in the two NSCLC cohorts. Antigen Processing and Presentation, Interleukins Receptor, Chemokine Receptor, Natural Killer Cell Cytotoxicity, TCR Signaling, and TGF-β family member receptors showed great predictive power (AUC > 0.7) by ROC curve analysis (Figure 6F). The scores of these IRPs were significantly higher in responders than in non-responders (Figure 6E). 

### 3.7. Identification of Potential Therapeutic Target Genes and Drugs Prediction

Herein, we applied a network-based guilt-by-association approach to calculate the proximity of each gene with dysregulated immune-related genes (IRGs) (see Materials and Methods). After integrating cancer-specific results (Appendix A), we obtained 13 genes (CDK2, DHX9, EEF1A1, EGFR, GRB2, IKBKE, MYC, SF3B3, SRC, TP53, TRAF6, YWHAG, YWHAZ) that scored in the top 1% among all cancer types. Then we re-performed a random walk with a restart with these 13 genes initially assigned the value “1”. As a result, our prediction identified 136 genes (top 1%), which had the strongest associations with the 13 initial genes (Figure 7A,B). Several IRGs involved in IRPs were also in the top 1% of our list. A module consisting of 12 genes was found in the protein-protein interaction network. Moreover, we aimed to identify potential therapeutic target genes and drugs; thus, we used the 136 genes as the query signatures and mapped them to the DGIdb database (Appendix A). 

As a result, we obtained a list of potential compounds and their target genes (shown in Table 1). AKT1 and EGFR have already been tested clinically for immune checkpoint blockade (ICB) therapies in cancers, which were initially involved in the above module [13]. Besides, some of the identified compounds were proven to have a clinical utility for ICB therapy, including TEMSIROLIMUS, BEVACIZUMAB, IBRUTINIB, and SUNITINIB [13,48]. Based on the observation, we speculated that other genes in the module might also be used as therapeutic targets. 

FYN was found to regulate autoimmunity by binding to B-lymphocyte surface antigen CD19 [49] and interacting with immunoglobulin superfamily member CD147 [50]. Evidence has shown that FYN plays a role in cancer pathogenesis and drug resistance [51]. Studies have shown the association between transcription factor JUN and drug resistance in several cancer types [52,53]; moreover, JUN could interact with the microenvironment, thus regulating inflammation and immunity [54,55,56,57]. The role of SRC in cancers has been well-established [58,59,60]; besides, up-regulated SRC kinase in the microenvironment could promote inflammation [61]. Numerous inhibitors have been developed to target SRC kinase [62]. The results might indicate that our prediction might have clinical implications for ICB therapeutic strategies. Moreover, we combined the cell line gene expression file and IC50 value of drugs from GDSC and CCLE to further validate our prediction. Results showed that the up- or down expression of drug targets could cause tumor cells to be resistant or sensitive to specific inhibitors (Figure 7D).

Functional enrichment analysis of the 136 genes showed that top-ranked genes were enriched in the initial 17 IRPs (BCR, TCR signaling, etc.) and other immune-related pathways; besides, several oncogene pathways were enriched, including Hippo, PI3K-AKT, WNT, etc. Moreover, aging- and longevity-related pathways, as well as drug-resist-related pathways, were also enriched in our prediction (Figure 7C).

## 4. Discussion

Accumulating evidence has linked immune-related pathways to cancer. However, what remains elusive is the effect of aging on immune-related pathways. Herein we comprehensively interrogated aging-related characteristics of IRPs and how these IRPs contribute to tumorigenesis. We found that tumors from related tissue origin showed a similar IRP expression pattern (Figure 1D). Most IRPs were dysregulated in tumors across 25 cancer types and the expression level changes showed a cancer-specific pattern. Specifically, we found that all IRPs were significantly up-regulated in testicular germ cell tumors. Several IRPs were significantly dysregulated with age in most cancer types. The strongest negative correlation between IRPs and age was found in testicular germ cell tumors. Specifically, we found the tumor- and age-related change of IRPs was in the opposite directions in LGG; the same condition was found in neurodegenerative diseases [63]. The previous study has shown connections between gliomas and neurodegenerative diseases [64]. Our conclusion that IRPs have aging-related characteristics was also validated by several aging benchmark genesets, including IMM-AGE, HAGR, and other GO genesets. Notably, we found the strongest positive correlation between TCR signaling pathways and conventional aging-related genesets (Figure 3) at both pan-cancer and cancer-specific levels. Previous studies have demonstrated the relationship between aging and TCR signaling in disease [65,66,67]. Collectively, our results demonstrated a complex relationship between immune-related pathways, aging, and cancer. Moreover, we also assessed the IRP differences between female and male. As a result, we found that IRPs showed the strongest gender-related differences in BRCA and THCA (Appendix A). Further study on gender-related IRPs in cancer might complement our study.

The prognostic value of immune-related genes (PD-1, PD-L1, CTLA-4, etc.) in cancers has been well-established [68,69,70,71,72]. Here, we extended our analysis to immune-related pathways and the results showed that at least one IRP had clinical associations in 17 cancer types. Specifically, based on our inspection of SKCM, all but TGF-β pathways could serve as prognostic markers in SKCM. Huge breakthroughs have been achieved in immunotherapies of melanoma [6,73,74]. Our results hint at the indispensable roles of immune-related pathways in cancer prognosis, especially in melanoma. Genetic alterations in oncogenic signaling pathways are typical hallmarks of cancer [75], which led us to evaluate the alterations in immune-related pathways. Of note, Cytokines and Cytokine Receptors pathways showed the highest frequency of alterations in pan-cancer, which were in line with a previous study [42]. Cytokines were proven essential in controlling host immune and inflammatory responses [76,77]. Increasingly studies have shown that cytokines are significant biomarkers in cancer and immunotherapy [78]. Our result suggested that genomic alterations in Cytokines pathways might serve as critical mediators in aging and cancer immunotherapy.

Unprecedented breakthroughs have been achieved in cancer treatment with the emergence of immune checkpoint blockade immunotherapy [79]. Durable benefits have been produced by inhibiting PD-1, PD-L1, CTLA-4, and other immunosuppressive checkpoint molecules [43]. Despite all the achievements, only a limited number of patients benefited from immune checkpoint blockade immunotherapy. Ongoing studies have been devoted to explaining the underlying mechanisms of adverse events after ICB immunotherapy. However, most current studies only focused on several essential checkpoint genes; the relationship between immune-related pathways and ICB responses has not received much attention. In the current study, up-regulated Antigen processing and presentation pathways in responders were found in melanoma and non-small cell lung cancer. In line with previous studies, members in the Antigen processing and presentation pathway could be a great signature in discriminating responders from non-responders [43,80]. Besides, the TCR signaling pathway also had a high discriminating power; a previously published in vivo study of mice supported this result [81,82]. Our findings revealed the indispensable roles of immune-related pathways in cancer immunotherapy. If more data are available in the future, our study is expected to be applied to more cancer patients who received immunotherapy.

Ongoing clinical tests have approved several drug combinations of checkpoint inhibitors, including ipilimumab [83,84,85,86], pembrolizumab, nivolumab [87,88], and atezolimumab [12]. Our study attempted to predict possible drug combinations for better immunotherapy response. We identified aging- and cancer-related genes by a random walk with a restart method in pan-cancer; using these136 genes as input, DGIdb helped suggest several potential drugs. Dasatinib is an inhibitor of SRC family kinases; targeting inhibition of SRC with dasatinib has been well-applied in various cancers [89,90,91,92,93,94]. Moreover, FYN has been validated to be one of the kinase targets of dasatinib [95,96,97]. Gemcitabine is a frontline agent in pancreatic cancer therapy [98], which could rewire immune-related pathways in the tumor microenvironment [99]. The combination of gemcitabine and anti-PD-1 enhanced the activity of M1 macrophages and CD8+ T cells in vivo [100,101]. Our result might provide more preclinical choices to help patients benefit from immunotherapy. Future experimental validation would improve our results and our work could provide potential candidates for future studies.

In this study, we used only 17 genesets from Immport, which curated immunologically relevant gene lists related to specific immune functions. In the future, applying more available datasets from other sources will further improve the research in related fields. Collectively, our systematic analysis of immune-related pathways in pan-cancer identified the aging-and tumor-related characteristics of these pathways and suggested that several pathways might help improve the response to cancer immunotherapy. Our results were instrumental in fully understanding the potential of immune-related pathways in cancer immunity and providing actionable strategies to broaden clinical tumor therapy.

## 5. Conclusions

In conclusion, our work established a link between aging, immunity, and cancers by applying transcriptomic and genomic analysis to immune-related genes and pathways. This study found a perturbation of expression and mutation pattern of immune pathways with aging characteristics in human cancer. These pathways could serve as potential biomarkers for cancer immunotherapy. Our work highlighted the importance of immune-related genes and pathways with oncogenic roles in the aging process, which could further provide insights into immunotherapy for patients.

## Figures and Tables

**Figure 1 cancers-15-00342-f001:**
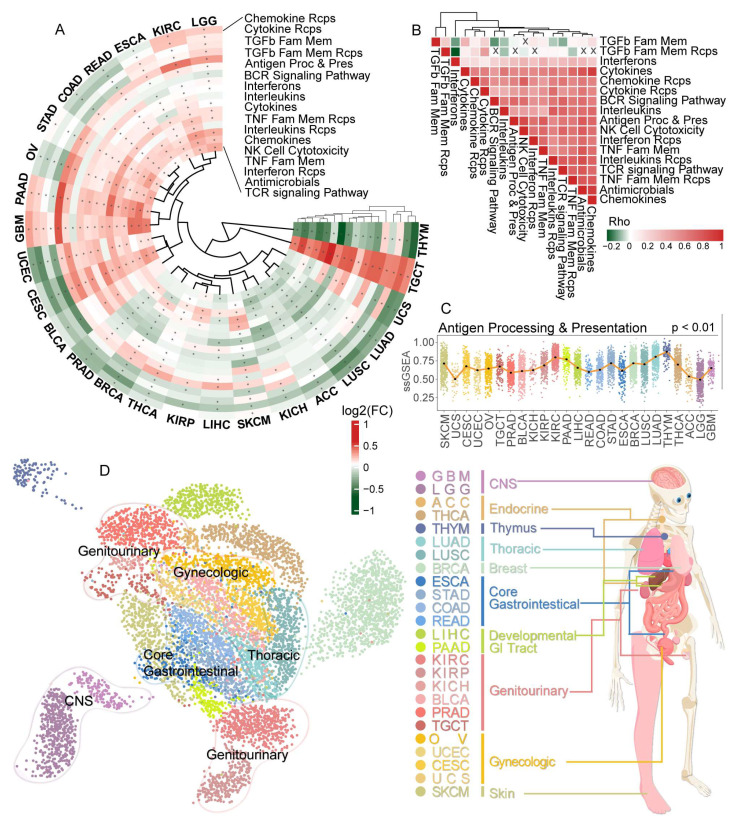
Dysregulated immune-related pathways in pan-cancer. (**A**) Circos heatmap showing the comparison of IRP expression level between tumor and normal across 25 cancer types. ‘*’ denotes a significant dysregulation. (**B**) Correlation heatmap of IRPs in pan-cancer. ‘X’ denotes no significant correlations. (**C**) Normalized IRP scores across 25 cancer types. (**D**) TCGA tumor samples clustered by Uniform Manifold Approximation and Projection (UMAP) analysis. Different colors denote various cancer types. Cancer types included: adrenocortical carcinoma (ACC), bladder urothelial carcinoma (BLCA), breast invasive carcinoma (BRCA), cervical squamous cell carcinoma and endocervical adenocarcinoma (CESC), colon adenocarcinoma (COAD), esophageal carcinoma (ESCA), glioblastoma multiforme (GBM), kidney chromophobe (KICH), kidney renal clear cell carcinoma (KIRC), kidney renal papillary cell carcinoma (KIRP), brain lower grade glioma (LGG), liver hepatocellular carcinoma (LIHC), lung adenocarcinoma (LUAD), lung squamous cell carcinoma (LUSC), ovarian serous cystadenocarcinoma (OV), pancreatic adenocarcinoma (PAAD), prostate adenocarcinoma (PRAD), rectum adenocarcinoma (READ), skin cutaneous melanoma (SKCM), stomach adenocarcinoma (STAD), testicular germ cell tumors (TGCT), thyroid carcinoma (THCA), thymoma (THYM), uterine corpus endometrial carcinoma (UCEC) and uterine carcinosarcoma (UCS).

**Figure 2 cancers-15-00342-f002:**
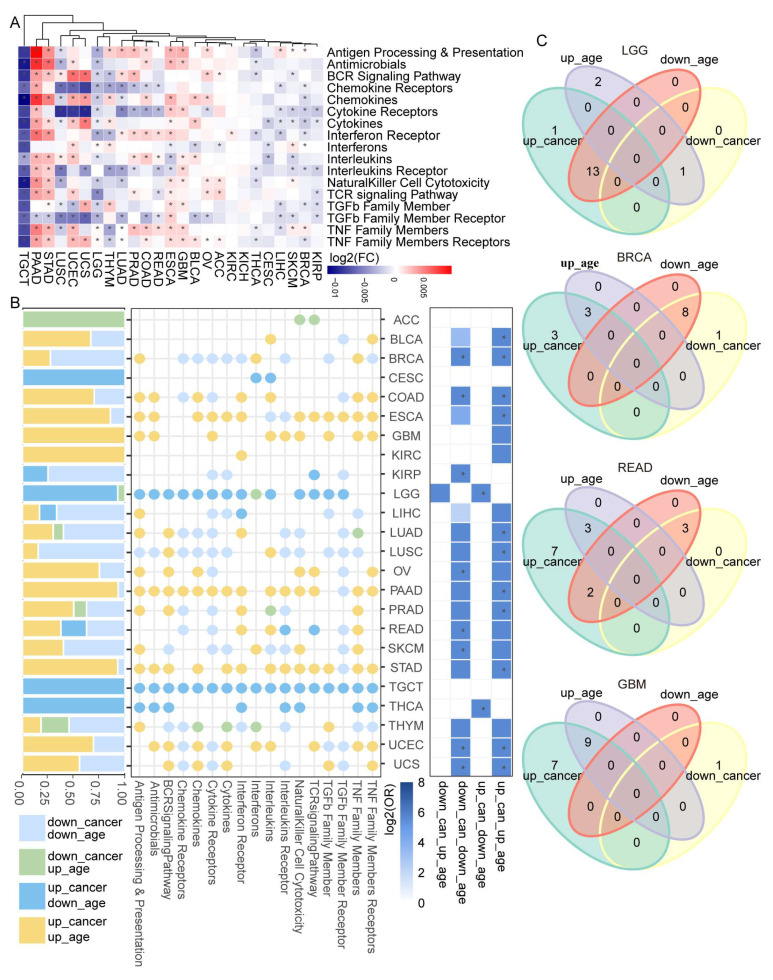
Immune-related pathways were dysregulated with age. (**A**) Heatmap showing IRP expression level dysregulated with age across 25 cancer types. ‘*’ denotes a significant dysregulation. (**B**) Overlap between dysregulated tumor- and age- IRPs. Left, stack bar showing the proportion of 4 types of IRPs across 25 cancer types. Middle, Dotplot showing the changing directions of 17 IRPs in 25 cancers. Right, Heatmap of Fisher’s exact test, colors denote the log2 (odds ratio) value. (**C**) Venn diagrams showing the overlap between dysregulated tumor- and age- IRPs.

**Figure 3 cancers-15-00342-f003:**
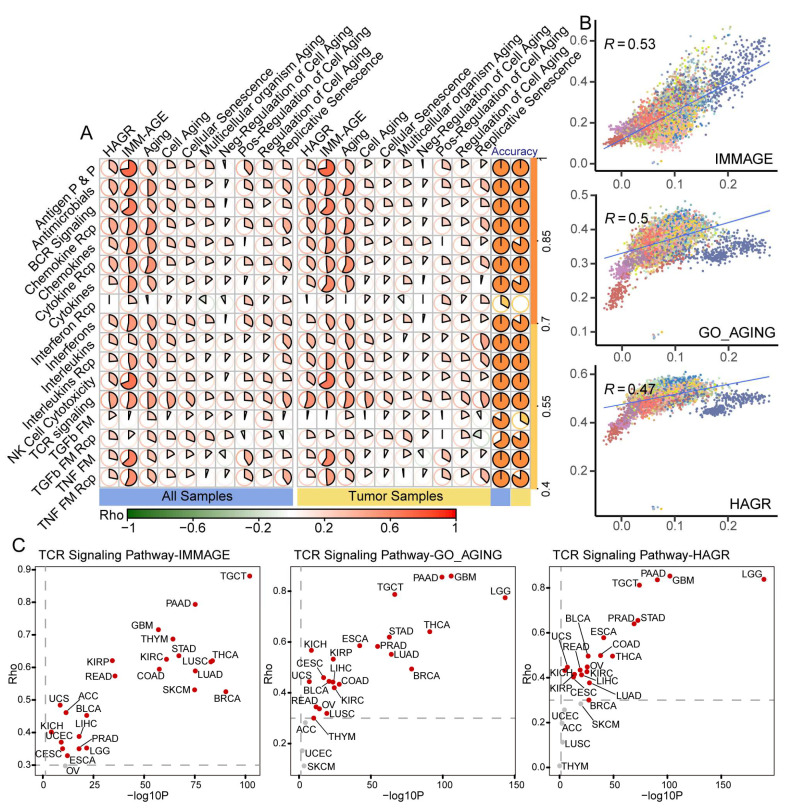
Evaluation of the age-related value of IRPs using benchmark gene sets. (**A**) Correlation pie plots between 17 IRPs and 10 ARGs in all/tumor samples. Right pie plots showing the validation results by mutual information analysis. (**B**) Correlation curves of TCR signaling pathways and 3 ARGs in all samples. Colors denote different primary tissue origins. (**C**) Volcano plots showing the correlations between TCR signaling pathways and 3 ARGs at cancer-specific level.

**Figure 4 cancers-15-00342-f004:**
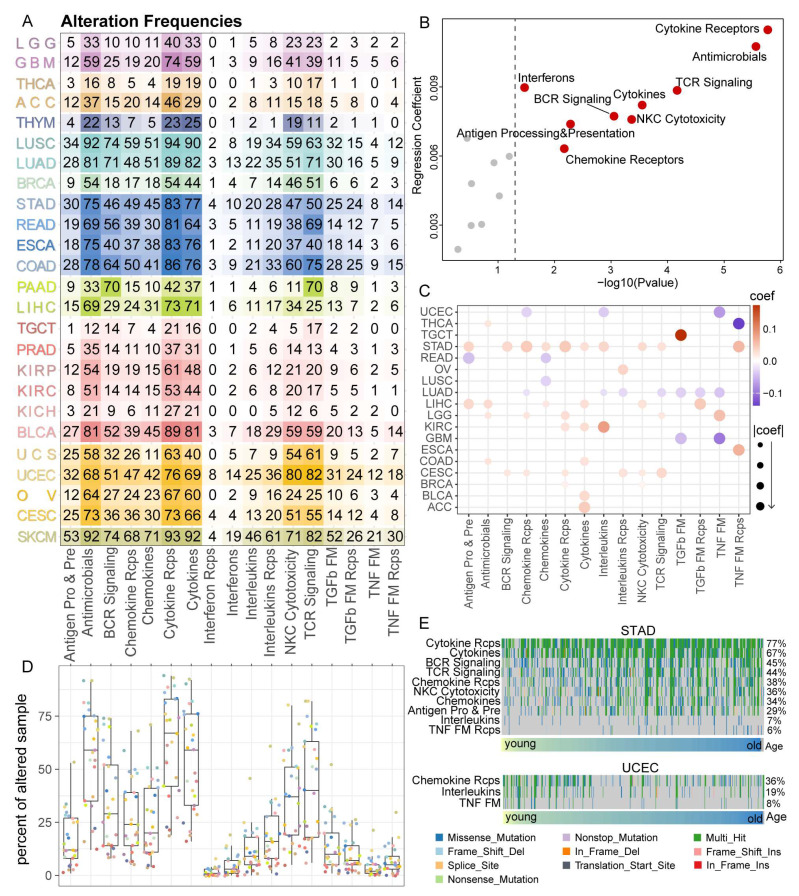
Alterations in immune-related pathways among 25 cancer types. (**A**) Heatmap showing the alteration frequencies of IRPs in various cancer samples, color intensities represent different percentages. (**B**) Volcano plots showing the associations between age and mutations in IRPs in pan-cancer. IRPs with significant positive correlations are highlighted in red. (**C**) Dotplot of associations between age and mutations in IRPs across various cancers. The color and size represent the logistic regression coefficients. (**D**) Boxplot showing the alteration frequencies of IRPs in various cancer samples. The color of dots denotes different cancer types (see Figure 1D). (**E**) Oncoplot of IRPs in STAD and UCEC.

**Figure 5 cancers-15-00342-f005:**
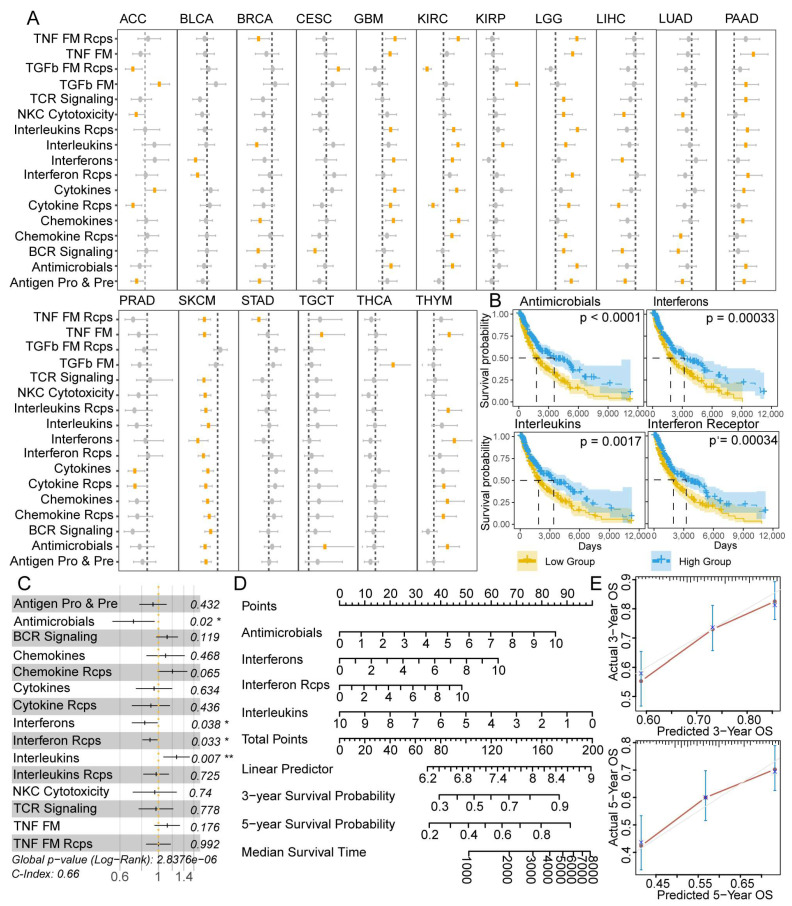
Clinical relevance of 17 IRPs among 25 cancer types. (**A**) Forest plot showing the results of Univariate Cox regression in pan-cancer. Rectangle dots represent hazard ratio (HR) and the error bars show the 95% confidence intervals of the HR. Orange denotes statistically significant. (**B**) KM curves for SKCM patients stratified by median level of 4 IRPs. (**C**) Forest plot showing the results of multivariate Cox regression in SKCM patients. ‘*’ denotes *p* < 0.05, ‘**’ denotes *p* < 0.01. (**D**) Nomogram using 4 IRPs for predicting the probability of 3- and 5-years OS. (**E**) Calibration plot evaluating of the predicted results by nomogram. The red and gray lines represent the difference between predicted and observed probability.

**Figure 6 cancers-15-00342-f006:**
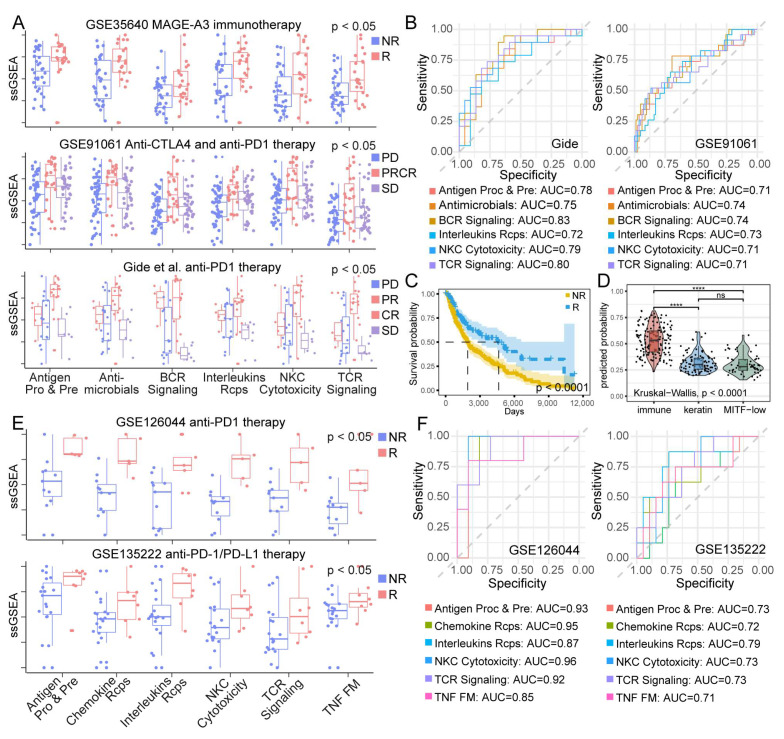
IRP scores were associated with patient response to immunotherapy. (**A**) Comparison of IRP expression level in melanoma patients between responders and non-responders. (**B**) ROC curves of 6 IRPs in 2cohorts of melanoma patients. (**C**) KM curve between predicted responder and non-responder in TCGA SKCM dataset. (**D**) Predicted possibility to be ICB the responder among three SKCM phenotype. ‘****’ denotes *p* < 0.0001, ‘ns’ denotes not significant. (**E**) Comparison of IRP expression level in lung cancer patients between responders and non-responders. (**F**) ROC curves of 6 IRPs in 2 cohorts of lung cancer patients.

**Figure 7 cancers-15-00342-f007:**
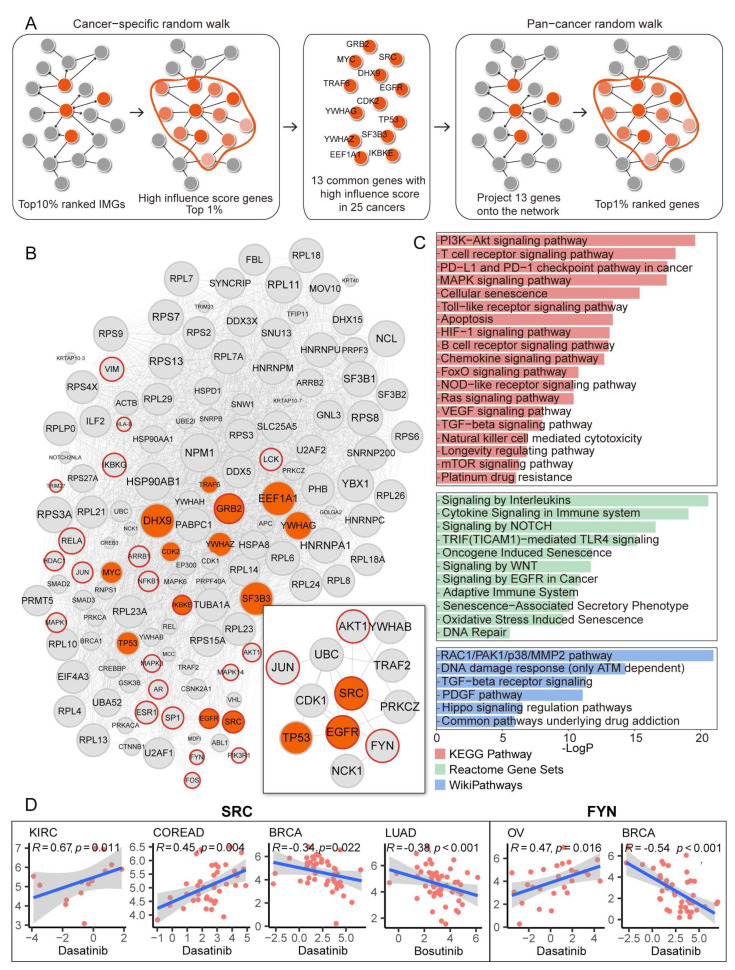
IRP scores were associated with patient response to immunotherapy. (**A**) Overview of identifying genes proximal to immune-related genes in pan-cancer in a PPI network. Seed genes are displayed in orange and projected onto a PPI network, followed by network propagation. (**B**) Protein–protein interaction network of top 1% genes in our prediction. The subnet in the box is the module identified by MCODE. (**C**) Bar plot showing the result of functional enrichment analysis, colors represent different data sources. (**D**) Correlation curves of the expression level of predicted drug targets and drugs in various cancer types. Dots represent the correlation between drug IC50 value and gene expression, and the blue line represents the curve fitting. The gray area represents the confidence interval.

**Table 1 cancers-15-00342-t001:** Predicted compounds and rank of the selected genes.

Symbol	Entrez ID	Rank in Our Prediction	Compound
AKT1	207	0.906%	TEMSIROLIMUS
TP53	7157	0.030%	BEVACIZUMAB
			BORTEZOMIB
EGFR	1956	0.097%	BEVACIZUMAB
			IBRUTINIB
			SUNITINIB
FYN	2534	0.178%	DASATINIB
JUN	3725	0.253%	COLCHICINE
SRC	6714	0.059%	DASATINIB
			BOSUTINIB
			GEMCITABINE

## Data Availability

The datasets supporting the conclusions of this article are available in the UCSC Xena repository, (http://xena.ucsc.edu/, accessed on 19 November 2020).

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
