# Peer review of "Immune Pathways with Aging Characteristics Improve Immunotherapy Benefits and Drug Prediction in Human Cancer"

_cancers, 2023, doi:10.3390/cancers15020342_

Round 1

Reviewer 1 Report

In the article entitled: “Immune Pathways with Aging Characteristics Improve Immunotherapy Benefits and Drug Prediction in Human Cancer.” the authors describe the interrelationship of immune-related pathways and patients' age characteristics by integrating transcriptomic and genomic pan-cancer data.

However, despite numerous valuable and very interesting data revealed, there are some obstacles that prevent the manuscript's publication in its current form.

These, among others, include:

The terms, such as IRP, SKCM, etc., should be followed by their full name in parentheses when mentioned for the first time in the manuscript text.

In the Materials and Methods section and subsection Data collection, the authors should indicate the date of the last accession to databases used for data collection. That also applies to the data obtained from the Molecular Signatures Database and Human Aging Genomic Resources. Also, the previously published studies (line 94) should be specified or referenced.

The material and method section should be rewritten in a more concise way  (also, some method descriptions are presented in the result section and missing in the materials and method section) and a little bit more understandable to readers outside the corresponding scientific field. Also, the corresponding papers describing the methods used should be referenced (seemingly not the case for all used analytical approaches) in the materials and method section. 

The analyzed age group stratification across the 25 cancer types analyzed should be specified/described. Was there any difference in patients' age groups across the  25 cancer types that could interfere with the overall data analysis? Were the non-cancerous tissues (corresponding data) available for all cancer types analyzed, and what kind of noncancerous tissue was analyzed in that tissue group- tissue from normal healthy individuals or tumor-adjacent tissue? Also, the gene expression profile of cancer cell lines i.e., the nature of cancer cell lines analyzed, should be better described/explained.

Did the authors analyze the corresponding IRPs (and their relationship with age stratification in pan-cancer data) with respect to the patient's gender subgroups? 

The meaning of IRPs  (1 and 2) mentioned in lines 82 and 85 is not completely clear or appropriately designated in the preceding sentence. 

Line 83-84 -- please define the adjusted p-value, i.e., the way of adjustment should be specified.

Line 119-120… Patients who were complete or partial were considered responders. -This sentence should be reformulated/explained.

The full names of the 25 cancer types analyzed in the manuscript should be mentioned at least in the first figure legend with corresponding abbreviations. Also, the synonyms for cancer types that are mentioned in the manuscript text (e.g.,  tumor tissues derived from the gastrointestinal tract, including ESCA, STAD, COAD, and READ) should be followed by their full name in parenthesis (when mentioned for the first time).

Line 206 -207… Several studies have collected essential genes associated with age. Here, we obtained ten gene sets from different sources to evaluate the aging-related value of 17 IRPs. - The corresponding references should be provided.

Table 1 is not mentioned in the manuscript text.

Line 401-402… Receptor pathways showed the highest frequency of alterations in pan-cancer, which were in line with a previous study. - the corresponding reference is missing.

A major revision is recommended.

Reviewer 2 Report

Title: Immune Pathways with Aging Characteristics Improve Immunotherapy Benefits and Drug Prediction in Human Cancer

General comments: Authors used transcriptomic and proteomic data and integrated these two to study and identified aging related immune pathways. It was a nice attempt and interesting data but need to clarify few important questions, see below:

  1. How many biologically defined gene sets were available in the initial database of ssGSEA software?
  2. What were the criteria used to calculate the IRP scores across 25 cancer types based on 17 ImmPort genesets? Why based on 17 ImmPort genesets, please provide a rationale and explain.
  3. What is the total number of genes in these 17 ImmPort genesets?
  4. Line 86 reads “It must be mentioned that the actual age of samples from GTEx was not provided”. Age of samples- does this mean it is the age of the person from whom the sample was collected or the age of the sample after collection from a patient? Please clarify.
  5. How was the noise in the measurements taken care of in the simulated data?
  6. Please provide the details on the type of tumors used for the generation g the transcriptomic/proteomic data.
  7. What cancer (25 types) types of data were used for analysis? Provide the information/list the cancer types.
  8. What cancers (organ-type cancers) and the similarities in gene expressions were identified?
  9. Did you find any unique signature of genes/data in any type of cancer? If so provide that information.
  10. As per, your results, might indicate that TCR and BCR signaling pathways could serve essential roles in aging and cancer, could you provide your analysis of why these pathways might have essential roles in aging and cancer?
  11. Line 299, read “Here, we collected five cohorts of SKCM patients (1 MAGE - A3 and 4 CTLA-4 and PD-1 blockade immunotherapy) and two cohorts of NSCLC”…here does cohort mean the group of patients, if so how many patients in each cohort?
  12. The results need to be verified by wet lab work using sample tumor cells that are sensitive and resistant to checkpoint inhibitor therapy.
  13. Why only in the melanoma and non-small cell lung cancer authors were able to find some correlation why not in other cancer types?

Round 2

Reviewer 1 Report

The authors have successfully revised the manuscript. I have no further comments or requests.

Reviewer 2 Report

Authors have responded to all of my queries satisfactorily in their capacity.